# The development and evaluation of a concussion education workshop for Gaelic games

Siobhán O'Connor *, Cliona Devaney, Enda Whyte, Aoife Burke

School of Health and Human Performance, Dublin City University, Dublin, Ireland

* siobhan.oconnor@dcu.ie

## Abstract

Concussions are frequent in Gaelic games and risky behaviours following a concussion are common. With the imminent integration of the Gaelic Athletic Association, Ladies Gaelic Football Association and Camogie Association, the development of a standardised concussion education initiative for all Gaelic games members is warranted. Thus, we aimed to develop a standardised concussion education workshop and evaluate if it improves concussion knowledge and attitudes in the Gaelic games community. A once-off concussion education workshop was developed in collaboration with the Gaelic games governing bodies and was delivered to 95 participants. Participants completed a survey (demographics, ROCKaS and the Perceptions of Concussion Inventory for Athletes [PCI-A]) pre-workshop and 1-month post-workshop (n = 55). Wilcoxon signed rank tests examined the differences pre- and 1-month post-workshop. One-month post-workshop, most participants strongly agreed/agreed that they can recognise concussion signs and symptoms (98.2%), know what to do in the event of a potential concussion (98.2%) and understand return to play guidelines (96.3%). Concussion knowledge (r = 0.34, p < 0.001), clarity (r = 0.45, p < 0.001) and control (r = 0.25, p = 0.01) significantly improved following the workshop. While concussion attitudes improved, the difference was not significant. No significant differences in anxiety, effects, treatment and symptom variability were noted from the PCI-A. A once-off time-efficient standardised concussion education workshop can enhance participants' concussion knowledge, clarity of concussion and beliefs of how much control they have over the outcomes of a concussion. A national rollout of the standardised concussion education workshop across the Gaelic games community, implemented as part of a wider concussion initiative, is recommended.

## 1. Introduction

Gaelic games are Ireland's national sports and include Gaelic football, Ladies Gaelic football, hurling and Camogie. They are governed by the Gaelic Athletic Association (GAA), Ladies Gaelic Football Association (LGFA) and the Camogie Association

**Data availability statement:** The raw data is available in an OSF repository at the following link: https://osf.io/nwfjy/?view_only=3873d40a6f854b258ae51d94ff1ef594.

**Funding:** This research project was funded by the 2023-2024 Sport Ireland Research Grant Scheme and the grant was applied for in collaboration with the Ladies Gaelic Football Association to fund the development and evaluation of the concussion education workshop. The funders had no role in study design, data collection and analysis, decision to publish, or preparation of the manuscript. No other conflicts of interest in the development and publication of this article are reported.

**Competing interests:** This research project was funded by the 2023-2024 Sport Ireland Research Grant Scheme and the grant was applied for in collaboration with the Ladies Gaelic Football Association to fund the development and evaluation of the concussion education workshop. No other conflicts of interest in the development and publication of this article are reported.

(CA) and are popular sports nationally, but also widely played globally, primarily by Irish communities [1,2]. Gaelic games are played at an amateur level, with competitions ranging from local club level to national inter-county level [3]. Gaelic games are fast-paced, contact field sports that require players to perform a wide range of high-intensity multidirectional movements including jumping, landing, rapid acceleration and deceleration, planting and cutting [4–6]. As a fast paced contact and/or non-incidental contact sport, concussions are of concern [7,8]. Concussions can lead to short-term neurological impairments and patients with concussion can present with an array of clinical signs and symptoms that can occur immediately or develop over the coming minutes, hours or days [9]. While sports-related concussions typically resolve in less than a month [10], persisting symptoms beyond this timeframe has been observed in 10–30% of patients [11].

Concussions are common in Gaelic games. Adult [12] and adolescent [8] Gaelic games players reported a lifetime prevalence of suspected or diagnosed concussion of 72.7% and 57.5% respectively. One-fifth of Ladies Gaelic footballers sustained a suspected or diagnosed concussion in one season [7]. In addition, most coaches (70.1%) and referees (74.5%) have had to manage a suspected concussion during Gaelic games [13]. While good overall knowledge of concussion has been previously observed in male and female Gaelic games adult players [12], adolescent players [8], coaches and referees [13], misunderstandings existed. Players, coaches and referees had difficulty recognising emotional symptoms of concussion and distinguishing false symptoms from accurate ones [8,12,13]. In addition, one-third of players were unaware that a graded return to play programme is required post-concussion [12]. Irish collegiate athletes (which included Gaelic games players) also expressed concerns regarding concussion [14] Those with a less clear understanding of concussion, displayed greater worry around concussions. Women, those without a diagnosed concussion in the past, those with a greater belief in the negative consequences of concussions and those who believed to a lesser extent that treatment can impact outcomes had greater anxiety around concussion.

Unsafe behaviours are also rife in Gaelic games following a suspected concussion, with non-disclosure of suspected concussions demonstrated previously. For example, in adult Gaelic games players, while almost three quarters reported a suspected concussion, just under 2 in 5 actually received a medical diagnosis of concussion [12]. Both adolescent and adult Gaelic games players were also less likely to report a concussion if there was a crucial game approaching [8,12]. Concussion guideline compliance is poor, with only 3.5% of concussed Ladies Gaelic football players following the entire LGFA concussion recommendations following a concussion [7]. In addition, in male inter-county Gaelic footballers and hurlers, while the majority of potential concussion events were assessed by medical professionals (87.2%, 86.3%), a limited number were then removed from play (5.0%, 7.1%) [15,16]. This is significant, as concussion mismanagement, such as continuing to play with a suspected concussion [17] and delayed attendance to a medical professional following concussion [18] can impact recovery. Typical of community sports in Ireland, regular access to medical personnel at matches and trainings is not common across

all codes of Gaelic games [8,12]. Consequently, the wider Gaelic games community, such as coaches, referees, fellow players and parents, are required to recognise a potential concussion if it occurs and manage the suspected concussion appropriately [8,12].

Recent research has noted that the wider Gaelic games community would welcome enhanced concussion education [8,12,13]. Previous concussion education interventions have been shown to improve concussion knowledge and attitudes [19]. However, it is essential that concussion education is tailored to the specific context it will be applied in [20]. Designing end-user focused concussion education initiatives and ensuring they are user-friendly, interactive and pitched at an appropriate level can enhance its chances of success [21,22]. Recent qualitative research, has highlighted the non-standardised concussion education available in Gaelic games [23] and recommended the development of regular education workshops across all members to enhance their concussion awareness [24]. In addition, previous engagement with current LGFA concussion education resources did not lead to safer concussion management in Ladies Gaelic footballers [7]. Thus, it is paramount that education initiatives are developed in partnership with governing bodies, and have a long-term implementation plan if initiatives' prove successful. Therefore, we aimed to develop a Gaelic games specific concussion education workshop and evaluate its ability to improve concussion knowledge and attitudes in male and female adult Gaelic games members.

## 2. Materials and methods

### 2.1. Study design and participants

A pre-post intervention study design was utilised on adult male and female Gaelic games members (such as players, coaches, referees etc). To be eligible, participants were required to be a current member of the GAA, LGFA and/or CA. Ethical approval was provided by the Dublin City University Research Ethics Committee (Approval number: DCUREC/2023/254).

### 2.2. Instrumentation

**2.2.1. Workshop.** The concussion education workshop was developed by 4 clinicians/researchers. All 4 were Certified Athletic Therapists and 3 were also active concussion researchers with extensive experience in Gaelic games concussion research across all codes. The 30-minute workshop included 4 sections. Section 1 explained what a concussion is, how common it is in Gaelic games, and detail on how currently it is not managed very well in Gaelic games. Section 2 detailed the signs and symptoms of a concussion to aid recognition of a concussion, highlighting how individual a concussion and recovery can be. Red flags were also highlighted and participants informed that any player experiencing these either immediately or after injury must be transferred to the hospital urgently. This section also included an interactive element where participants were asked if they had ever dealt with a Gaelic games player with a concussion and how they presented when it happened. Section 3 provided a flowchart that guided Gaelic games members on what to do if they suspect a player has a concussion. This detailed the importance of recognition and removal, essential dos and don'ts for immediate management, and what can potentially make a concussion worse (e.g., continuing to play). It also highlighted the next steps following recognition and removal, including the importance of exercise, re-evaluation from a healthcare professional and the role healthcare professionals can provide to help concussion recovery. It also provided an overview of return to learn/work and return to sport strategies for Gaelic games and what to do if someone has persisting concussion symptoms. This section also included an interactive component where participants that experienced a concussion before or knew someone who had, discussed their rehabilitation and return to play if comfortable to do so. Section 4 then provided a summary of the workshop and key take home messages for Gaelic games members to consider. Supplementary material 1 details the workshop's script. Embedded in the workshop was a video of a current inter-county Gaelic games player's recent lived experience of a concussion. Regular input from representatives of the GAA, LGFA and CA was provided to ensure the workshop abided by all concussion guidance across codes and was

focused on the needs of the organisations and their members. Five meetings were held with representatives from all organisations during the development of the workshop and additional meetings with individual organisations was also held as requested (n = 6). The concussion education workshop was primarily developed using existing best practice recommendations [9]. the GAA, LGFA and CA concussion management guidelines and recent concussion research in Gaelic games [7,8,12,13,25]. We also ensured the workshop content included all recommended topics proposed by the NCAA and Department of Defence Delphi consensus on concussion education [21]. The workshop was piloted both online (n = 14) and in-person (n = 32) to ensure the content, pace, and visuals were pitched appropriately.

**2.2.2. Survey.** The survey was comprised of three sections. Section 1 included 2 screening questions to determine eligibility (confirmation they were over 18 and a current member of the GAA, LGFA or CA). Participants then detailed their name, email address, age and gender. Section 2 included the valid and reliable Rosenbaum Concussion Knowledge and Attitudes Survey (RoCKAS) which measures concussion knowledge and attitudes [26]. Two scales were created. The Concussion Knowledge Index (CKI) included 25 questions (scoring range 0–25) with correct responses provided a score of one and incorrect responses scored a zero. The Concussion Attitude Index (CAI) included fifteen concussion scenarios where participants rated their level of agreement on a five-point Likert scale (strongly disagree to strongly agree). Responses were summed from 1–5 with a higher score indicating better concussion attitudes (scoring range 15–75). A validity scale is also included, and participants that display scores of less than 2 in these suggests respondents are not responding to the survey with appropriate effort and their responses should be removed. Acceptable reliability was noted for CAI (Cronbach's alpha = 0.82). Finally, Section 3 consisted of the Perceptions of Concussion Inventory for Athletes (PCI-A) [27] whereby participants rated their agreement to 21 statements on a 5-point Likert scale. Six outcome scales were then developed namely anxiety (4 items, scoring range 4–20), effects (4 items, scoring range 4–20) control (3 items, scoring range 3–15) clarity (4 items, scoring range 40–20) treatment (3 items, scoring range 3–15), and symptom variability (3 items, scoring range 3–15). Acceptable internal consistency was observed for the current study (Cronbach's alpha = 0.79). Participants in the post-intervention survey provided their name, their agreement on a 5-point Likert scale (strongly disagree to strongly agree) to 7 statements regarding their concussion understanding following the workshop and repeated the RoCKAS and PCI-A scales.

## 2.3. Procedures

Participants were recruited via an email sent to clubs to distribute to their members, social media and word of mouth. Data collection occurred from 17th July 2024–8th of November 2024. Using G*Power sample size calculator a minimum of 45 participants were required to attain an alpha volume of 0.05 and power of 0.80. Prior to the workshop, participants provided written informed consent, followed by the survey which was distributed via email. The concussion education workshop which took on average 25 minutes but inclusive of questions ranged from 30–50 minutes, was then delivered both online and in-person by a Certified Athletic Therapist with clinical expertise in concussion recognition and management. Ten workshops took place across 6 counties (and 3 provinces) to 95 participants (6–18 participants in each workshop). One month following completion the survey was emailed to all members to complete again. Two reminders were then sent. Sixty-five participants completed the post-workshop survey one month following the workshop. However, 10 were removed due to incomplete responses (did not fully complete any scale). Thus, 55 participants were included in the final analysis.

## 2.4. Statistical analysis

Survey data was exported from Qualtrics XM (Qualtrics XM, USA) to Microsoft Excel (Version 16.88, Microsoft, USA) and participants were allocated a unique identification number to ensure anonymity in survey analyses. Concussion Knowledge Index (CKI) scores and Concussion Attitude Index (CAI) scores were coded in Excel to give a total ROCKaS score. The Perceptions of Concussion Inventory for Athletes (PCI-A) scores were also coded and totalled in Excel. The

coded data was imported to SPSS (IBM SPSS, Version 29, USA), for statistical analyses. The data were non-normally distributed, so Wilcoxon Signed Rank Tests were used to assess pre- and post-workshop differences for 1. CKI, 2 CAI, 3. PCIA-A Anxiety, 4. PCI-A Effects, 5. PCI-A Control, 6. PCI-A Clarity, 7. PCI-A Treatment and 8. PCI-A Symptom Variability scores. An alpha level of $p < 0.05$ was interpreted as being significant. Effect sizes were calculated, with the following classification according to Cohen [28]: $r = 0.1$ considered as small; $r = 0.3$ considered as medium and $r = 0.5$ considered as large.

## 3. Results

Slightly more men (54.5%, $n = 30$) than women completed the data collection (45.5%, $n = 25$). Participants had a mean age of $43.8 \pm 10.6$ years. Table 1 displays their understanding of concussion post-workshop, with high percentages strongly agreeing/agreeing that they recognise the main signs and symptoms of a concussion (98.2%, 54), know what to do if they think a player has concussion (98.2%, 54) and understand the return to play guidelines (96.3%, 53).

The mean and standard deviation scores from the ROCKaS and PCI-A pre- and post-workshop are displayed in Table 2. There was a significant improvement in CKI scores from pre-workshop ($19.7 \pm 1.4$, MD: 20) to post-workshop ($21.0 \pm 2.1$, Md: 21), with a medium effect size ($r = 0.34$, $p < 0.001$). Concussion attitude scores also improved from pre- ($65.8 \pm 6.9$, MD: 67) to post-workshop ($67.6 \pm 5.5$, MD: 69), with a small effect size ($r = 0.24$), but this was not significant ($p = 0.13$). While no significant differences for anxiety, effects, treatment and symptom variability was observed, the clarity PCI-A subscale significantly improved with a large effect size ($14.5 \pm 2.6$, MD: 14 *vs* $16.7 \pm 1.7$, MD: 16, $r = 0.45$, $p < 0.001$). In addition, the scores for the control subscale were also significantly higher 1-month post-workshop ($13.4 \pm 1.4$, Md: 13) than per-workshop ($12.4 \pm 1.9$, Md: 12, $p = 0.01$), with a small effect size ($r = 0.25$).

**Table 1. Self-perceived concussion understandings 1-month post-workshop survey ($n = 55$).**

| Statement | Strongly Agree % (n) | Agree % (n) | Neutral % (n) | Disagree % (n) | Strongly Disagree % (n) |
|---|---|---|---|---|---|
| I understand what a concussion is | 60.0 (33) | 38.2 (21) | 0.0 (0) | 0.0 (0) | 1.8 (1) |
| I recognise the main signs and symptoms of concussion | 45.5 (25) | 52.7 (29) | 0.0 (0) | 0.0 (0) | 1.8 (1) |
| I know what to do if they think they or a player has a concussion | 34.5 (19) | 63.6 (35) | 0.0 (0) | 0.0 (0) | 1.8 (1) |
| I understand the return to play guidelines | 34.5 (19) | 61.8 (34) | 1.8 (1) | 0.0 (0) | 1.8 (1) |
| I understand the possible long-term effects of concussion | 40.0 (22) | 56.4 (31) | 1.8 (1) | 0.0 (0) | 1.8 (1) |
| I understand the effects of poor concussion management | 47.3 (26) | 50.9 (28) | 0.0 (0) | 0.0 (0) | 1.8 (1) |
| I know the main risk factors of concussion | 30.9 (17) | 63.6 (35) | 3.6 (2) | 0.0 (0) | 1.8 (1) |

**Table 2. Pre- and 1 month post-workshop CKI, CAI and PCI-A scores.**

| Survey component | Pre-workshop | Post-workshop | P value | Effect size (r) |
|---|---|---|---|---|
| CKI | $65.7 \pm 7.0$ | $67.2 \pm 5.6$ | <0.0001* | 0.34 |
| CAI | $65.8 \pm 6.9$ | $67.6 \pm 5.5$ | 0.13 | 0.15 |
| PCI-A Anxiety | $13.2 \pm 2.4$ | $13.4 \pm 3.2$ | 0.78 | 0.03 |
| PCI-A Effects | $11.1 \pm 2.3$ | $10.4 \pm 2.3$ | 0.10 | 0.16 |
| PCI-A Control | $12.4 \pm 1.9$ | $13.4 \pm 1.4$ | 0.01* | 0.25 |
| PCI-A Clarity | $14.5 \pm 2.6$ | $16.7 \pm 1.7$ | <0.001* | 0.45 |
| PCI-A Treatment | $10.5 \pm 1.8$ | $11.7 \pm 1.9$ | 0.11 | 0.16 |
| PCI-A Symptom | $10.5 \pm 1.8$ | $10.6 \pm 2.0$ | 0.84 | 0.02 |

CKI: Concussion Knowledge Index; CAI: Concussion Attitude Index; PCI-A: Perceptions of Concussion Inventory for Athletes.

## 4. Discussion

This study aimed to examine whether a concussion education workshop can improve concussion knowledge and attitudes among Gaelic games members. All Gaelic games members, including players, coaches, referees and parents of underage players, frequently experience having to deal with potentially concussive events [7,8,12,13]. Thus, improving the wider Gaelic games community's understanding of concussions, how to manage them appropriately and evidence-based treatment pathways is critical to optimise how we deal with suspected concussions in the community sports of Gaelic games [29]. With the GAA, LGFA and CA committing to full integration of their organisations by 2027, and the future move to a "one club" model, it is imperative to develop and implement standardised concussion education across all codes of Gaelic games. Overall, participants found this once-off short and cost-effective concussion education intervention valuable. The vast majority of participants after the workshop felt they understood what a concussion is, recognised the main signs and symptoms, knew what to do in the event of a suspected concussion and understood the Gaelic games return to play guidelines. Concussion education should be specific to the needs of the recipients and concussion messaging should include role models or real stories from those who have experienced a concussion and reflect the cultural, sporting and healthcare context it takes place in [20,30,31]. This concussion education workshop aimed to address these factors by incorporating the player's voice and lived concussion experiences, alongside specific messaging addressing previously identified concussion misconceptions and highlighting the appropriate concussion referral route in the Irish healthcare space. Partnering with governing bodies and decision makers within organisations that can truly enact change within the sports, as we did in this study, to develop and deliver concussion education is an important step to facilitate widespread dissemination of the educational strategy [21,32].

Similar to previous concussion education initiatives [19], we found our one-off concussion education workshop significantly enhanced concussion knowledge. In addition, participants' clarity or understanding of a concussion and their symptoms also significantly improved. Standardised concussion education can provide high quality and accurate messaging to enhance concussion awareness and address common concussion misconceptions within the sporting community. In an Irish context this is very important, as previous research in Irish collegiate athletes reported one quarter had never received any concussion information, and potentially unreliable sources were frequently used to gather concussion information (e.g., movies and books, social media, news outlets, online sources) [30]. Thus, the national rollout of this concussion education workshop to enhance concussion knowledge and awareness of appropriate management and return to sport is recommended across all three Gaelic games organisations. The greatest improvements in concussion knowledge are observed immediately after interventions occur, and decline over time [33]. In addition, it is important to note that high levels of concussion knowledge alone do not lead to safer concussion behaviours [12,34]. Thus, regular mandatory education that is part of a larger concussion initiative is recommended [19,31]. Control, or their beliefs that their actions can impact concussion outcomes was also significantly improved 1-month post workshop. Research has found that risky behaviour post-concussion, such as continuing to play on and lack of engagement with medical professionals for assessment and treatment, can negatively impact recovery [17,18]. Thus, the recognition that players' actions can directly impact their recovery could potentially enhance the likelihood to engage in safer behaviour following a suspected concussion. In fact, higher control has been associated with better psychological well-being, social functioning, utilisation of more problem focused coping strategies, expression of emotions and negatively related to psychological distress [35].

While concussion attitudes were higher 1-month following the workshop (CAI: 65.8 *vs* 67.6), the improvement was not statistically significant. A recent systematic review found that while most concussion education initiatives improved concussion knowledge, attitudes were less commonly enhanced [19]. Favourable concussion attitudes have been shown to be one factor that can positively improve the likelihood of concussion disclosure [29]. However, concussion knowledge, stigma, pressure and whether they perceive the environment as supportive can also influence their decision-making [29,36]. No significant improvements in anxiety regarding concussions, consequence of a concussion, beliefs regarding

treatment or symptom variability were found. Unfortunately, concussion education alone is unlikely to fundamentally change concussion culture [21,36]. All Gaelic games club members, and the policies and procedures within the three organisations and individual clubs, should positively endorse the concussion education messaging provided in the workshop and strive to create a supportive and open concussion culture and environment conducive to concussion reporting within their teams and club [21,29,32]. Coaches, and all role models within clubs, should also explicitly encourage concussion reporting and highlight the importance of taking the time to treat the injury appropriately to minimise the intrapersonal pressure players may feel to compete while experiencing symptoms [21,29].

The short follow up duration of one month is a substantial limitation to this study. Future research should examine the impact across a season. Social desirability may have also been an issue, with participants potentially providing answers that they believe the research team would like, rather than their true views. Using self-report scales to examine knowledge and attitudes may lack the nuance required to truly unpack the specific knowledge or attitude improvements that are observed. Thus, future research should qualitatively examine the views of participants on the concussion education workshop. Finally, evaluation of whether the national rollout of a workshop of this nature impacts practices following a concussion is recommended.

## 5. Conclusions

The standardisation of concussion education across all Gaelic games' codes, in line with future integration plans, is an important step in enhancing the recognition and management of concussions in Gaelic games. A once-off time and cost-efficient concussion education workshop significantly improved Gaelic games members concussion knowledge, their clarity or understanding of a concussion and their perceptions of how much control they have over the outcomes of a sustained concussion. Thus, rolling out this workshop nationally to the wider Gaelic games community is an important step to enhance concussion recognition and support appropriate concussion management across Gaelic games. While, attitudes did improve, this was not significant, and no significant enhancements in their perceptions of concussion regarding anxiety, effects, treatment and symptom variability was observed. Thus, future research should examine how to truly address concussion culture in Gaelic games, and the development of a wider concussion initiative, inclusive of the concussion education workshop should be considered.

## Author contributions

**Conceptualization:** Siobhán O'Connor, Enda Whyte, Aoife Burke.

**Formal analysis:** Siobhán O'Connor, Aoife Burke.

**Funding acquisition:** Siobhán O'Connor.

**Investigation:** Cliona Devaney.

**Methodology:** Siobhán O'Connor, Cliona Devaney.

**Project administration:** Siobhán O'Connor, Cliona Devaney, Enda Whyte, Aoife Burke.

**Supervision:** Siobhán O'Connor, Enda Whyte, Aoife Burke.

**Writing – original draft:** Siobhán O'Connor.

**Writing – review & editing:** Cliona Devaney, Enda Whyte, Aoife Burke.

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
