## [Decision Letter · Decision Letter 0]

The Development and Evaluation of a Concussion Education Workshop for Gaelic games

PONE-D-25-18087

Dear Dr. O'Connor,

We’re pleased to inform you that your manuscript has been judged scientifically suitable for publication and will be formally accepted for publication once it meets all outstanding technical requirements.

Kind regards,

Julio Alejandro Henriques Castro da Costa

Academic Editor

PLOS ONE

This research project was funded by the 2023-2024 Sport Ireland Research Grant Scheme.

Please respond by return e-mail so that we can amend your financial disclosure and competing interests on your behalf.

4. We are unable to open your Supporting Information file [SPSS file for Concussion Education Workshop Date Respository.zip]. Please kindly revise as necessary and re-upload.

Reviewers' comments:

Reviewer's Responses to Questions

**Comments to the Author**

1. Is the manuscript technically sound, and do the data support the conclusions?

Reviewer #1: Yes

Reviewer #2: Yes

2. Has the statistical analysis been performed appropriately and rigorously?

Reviewer #1: Yes

Reviewer #2: Yes

3. Have the authors made all data underlying the findings in their manuscript fully available?

Reviewer #1: Yes

Reviewer #2: Yes

4. Is the manuscript presented in an intelligible fashion and written in standard English?

Reviewer #1: Yes

Reviewer #2: Yes

Reviewer #1: In the context of Gaelic sports, the possibility of suffering a concussion is high. Evaluating the effectiveness of educational workshops focused on the treatment and management of these injuries is not only appropriate but crucial for the health and safety of players, especially in amateur settings where access to specialized medical personnel is limited.

The study is based on principles of health pedagogy and the sports injury prevention model. Lack of knowledge about the symptoms, recovery time, and consequences of a concussion can lead to an athlete's premature return to play, increasing the risk. Thus, education emerges as a powerful tool to change attitudes, improve early recognition of clinical signs, and encourage injury reporting by athletes and coaches. The findings allow us to reflect on participants' prior awareness of the severity of concussions, as well as the workshop's effectiveness in improving their knowledge and willingness to respond appropriately to an injury.

The study validates a contextualized educational tool and contributes to a safer sports culture.

Reviewer #2: authors have presented the Research methodology appropriately. Results, discussion and conclusion has also been explained appropriately on the basis of data. data supports conclusion. Discussion part is presented in connection to Research problem

**Do you want your identity to be public for this peer review?** For information about this choice, including consent withdrawal, please see our Privacy Policy

Reviewer #1: **Yes: ** Elkin Alberto Arias Arias

Reviewer #2: No

---

## [Editor Report · Acceptance letter]

PONE-D-25-18087

PLOS ONE

Dear Dr. O'Connor,

I'm pleased to inform you that your manuscript has been deemed suitable for publication in PLOS ONE. Congratulations! Your manuscript is now being handed over to our production team.

Kind regards,

on behalf of

Dr. Julio Alejandro Henriques Castro da Costa

Academic Editor

PLOS ONE